# Update on Diabetic Kidney Disease (DKD): Focus on Non-Albuminuric DKD and Cardiovascular Risk

**DOI:** 10.3390/biom13050752

**Published:** 2023-04-26

**Authors:** Sabrina Scilletta, Maurizio Di Marco, Nicoletta Miano, Agnese Filippello, Stefania Di Mauro, Alessandra Scamporrino, Marco Musmeci, Giuseppe Coppolino, Francesco Di Giacomo Barbagallo, Giosiana Bosco, Roberto Scicali, Salvatore Piro, Francesco Purrello, Antonino Di Pino

**Affiliations:** Department of Clinical and Experimental Medicine, University of Catania, 95122 Catania, Italy; sabrinascilletta@gmail.com (S.S.); maurizio.dimarco@studium.unict.it (M.D.M.); nicoletta.miano@gmail.com (N.M.); agnese.filippello@gmail.com (A.F.); 8stefaniadimauro6@gmail.com (S.D.M.); alessandraska@hotmail.com (A.S.); marco.musmeci96@gmail.com (M.M.); giuseppecoppolino93@gmail.com (G.C.); fdigiacomobarbagallo@gmail.com (F.D.G.B.); giosiana.bosco@gmail.com (G.B.); roberto.scicali@unict.it (R.S.); salvatore.piro@unict.it (S.P.); francesco.purrello@unict.it (F.P.)

**Keywords:** chronic complications of type 2 diabetes, diabetic kidney disease, cardiovascular risk, non-albuminuric diabetic kidney disease

## Abstract

The classic description of diabetic kidney disease (DKD) involves progressive stages of glomerular hyperfiltration, microalbuminuria, proteinuria, and a decline in the estimated glomerular filtration rate (eGFR), leading to dialysis. In recent years, this concept has been increasingly challenged as evidence suggests that DKD presents more heterogeneously. Large studies have revealed that eGFR decline may also occur independently from the development of albuminuria. This concept led to the identification of a new DKD phenotype: non-albuminuric DKD (eGFR < 60 mL/min/1.73 m^2^, absence of albuminuria), whose pathogenesis is still unknown. However, various hypotheses have been formulated, the most likely of which is the acute kidney injury-to-chronic kidney disease (CKD) transition, with prevalent tubular, rather than glomerular, damage (typically described in albuminuric DKD). Moreover, it is still debated which phenotype is associated with a higher cardiovascular risk, due to contrasting results available in the literature. Finally, much evidence has accumulated on the various classes of drugs with beneficial effects on DKD; however, there is a lack of studies analyzing the different effects of drugs on the various phenotypes of DKD. For this reason, there are still no specific guidelines for therapy in one phenotype rather than the other, generically referring to diabetic patients with CKD.

## 1. Introduction

Diabetic kidney disease (DKD) is a common microvascular complication that develops in approximately 40% of patients with diabetes. It is the main cause of chronic kidney disease (CKD) and end-stage renal disease (ESRD) and is a major cause of mortality and morbidity in diabetics not only due to ESRD but also because of the resulting cardiovascular risk [1].

The clinical relevance of DKD compared with other complications of type 2 diabetes needs to be highlighted; indeed, while other complications associated with diabetes have seen a decline in recent years, the same has not occurred for DKD: possible explanations for this phenomenon are increased ESRD survival with the improvement of cardiovascular prognosis or the still unmet need for better treatment [2].

## 2. Non-Albuminuric Phenotype: Clinical Presentation, Epidemiology and Risk Factors

### 2.1. Clinical Presentation

The classic description of DKD involves progressive stages of glomerular hyperfiltration, microalbuminuria, overt proteinuria, and a decline in the estimated glomerular filtration rate (eGFR), eventually leading to dialysis [3,4]. In the last few years, this concept has been increasingly challenged as evidence suggests that DKD in the contemporary era presents in a more heterogeneous manner. Large cross-sectional studies reveal that a large group of patients with type 2 diabetes and reduced kidney function presents without albuminuria, suggesting that both the onset and progression of the decline in renal function may also occur independently from the development of albuminuria. This concept led to the identification of a new DKD phenotype: non-albuminuric DKD [5].

### 2.2. Epidemiology

Studies documented that non-albuminuric DKD (eGFR < 60 mL/min/1.73 m^2^ in the absence of albuminuria) occurs relatively frequently in patients with diabetes and its prevalence is increasing. In the National Health And Nutrition Examination Survey (NHANES) from 1988 to 1994 [6], 20% of subjects with diabetes had advanced kidney disease (eGFR < 30 mL/min/1.72 m^2^) without the presence of albuminuria; furthermore, 30% of participants 40 years and older with type 2 diabetes and low eGFR had no albuminuria or retinopathy. The authors of the NHANES hypothesized that this dissociation between kidney disease and other microvascular complications may be due to some pathologic lesions that were different from typical histological diabetic lesions, such as glomerulosclerosis. In more recent years, the National Evaluation of the Frequency of Renal Impairment Coexisting with NIDDM (NEFRON) survey of primary care patients with type 2 diabetes found that 55% of those with low eGFR were persistently non-albuminuric [7].

In fact, in the last few decades, the prevalence of DKD has not decreased due to inverse changes in the two main manifestations of DKD: albuminuria (urine albumin-to-creatinine ratio, UACR ≥30 mg/g), whose prevalence decreased, and low eGFR, whose prevalence increased [8]. This variation in the disease course might be due to changes in the prevalence of comorbidities, such as an increase in the prevalence of hypertension and obesity within an ageing population, a reduction in the prevalence of smoking, increased use of multifactorial interventions (leading to improved glucose, blood pressure, and lipid management), and the use of certain agents (such as anti-hypertensive renin–angiotensin–aldosterone system [RAAS] inhibitors and sodium/glucose co-transporter 2 [SGLT2] inhibitors) [9] with demonstrated effects on cardiovascular mortality and reduction of the progression of kidney disease [10]. The rising prevalence of the non-albuminuric phenotype could be explained by ongoing RAAS regulation, which decreases albuminuria by altering renal hemodynamics [11], although other renal pathophysiological mechanisms may also be involved.

### 2.3. Risk Factors

Afghahi et al. examined the clinical characteristics and prevalence of non-albuminuric DKD in the Swedish National Diabetes Register [12]. In this study, 28% of diabetic patients with non-albuminuric DKD were not receiving treatment with RAAS blocking agents and such treatment was not a predictor of non-albuminuric DKD. Furthermore, patients with DKD and albuminuria were more likely to develop retinopathy (31%). The association between several risk factors and the non-albuminuric DKD phenotype has been observed in several studies. In the Renal Insufficiency And Cardiovascular Events (RIACE) cohort [13], non-albuminuric DKD patients were more frequently female and nonsmokers, and had lower levels of glycated hemoglobin (HbA1c), but did not have longer diabetes duration compared with those patients with albuminuria. Multiple regression analysis confirmed that the non-albuminuric phenotype is associated with women, but not with HbA1c.

Accordingly, in a cohort of 562 Korean patients with type 2 diabetes [14], the normoalbuminuric DKD phenotype was associated with women, a shorter duration of diabetes, a lower prevalence of diabetic retinopathy, and a lower prevalence of antihypertensive medications use when compared with micro- and macro-albuminuric kidney disease. Therefore, non-albuminuric DKD decreased progressively with an increase in the duration of diabetes and an increase in the severity of retinopathy. These clinical characteristics were largely similar to those seen in other studies.

A report concerning 660 patients with type 2 diabetes and normoalbuminuria [15] showed that subjects with reduced eGFR had higher levels of insulin resistance, total and LDL cholesterol, and triglycerides as well as a higher prevalence of the metabolic syndrome compared with those with preserved eGFR.

All data are shown in Figure 1 and suggest that the pathogenesis and progression of the non-albuminuric form may follow a distinct pathway from the albuminuric one [16].

## 3. Pathogenesis and Histopathology

### 3.1. Histopathology

Several pathophysiologic pathways [17] are involved in the development of classic DKD, which can be summarized as metabolic, hemodynamic [18], and inflammatory, with the role of many growth, proinflammatory [19,20], or profibrotic factors [21,22].

These pathways, driven by hyperglycemia, result in pathological damage to the glomerulus, particularly in podocytes, and the tubulointerstitium, which leads to an increase in glomerular albumin permeability (albuminuria) and subsequent reductions in eGFR. According to an international consensus conference, the histological manifestations of diabetic nephropathy follow four progressive classes [23] (Table 1). The earliest change that occurs in the kidney is a thickening of the glomerular basement membrane (GBM) (class I) [24]: light microscopy shows minimal, non-specific, or no changes. Thickening of the GBM does not directly correlate with clinical injury. Patients may have such thickening but have no increase in their urine albumin excretion rate or impairment of eGFR in this stage. Class II is characterized by mesangial expansion [25]. In particular, class IIa is characterized by mild expansion in >25% of the observed mesangium, while class IIb is characterized by severe expansion. An increase in mesangial matrix, glomeruli, and kidney volume is clinically manifested as kidney enlargement. Urine albumin excretion is often increased in these patients. An increase in the mesangial matrix is followed by mesangial sclerosis (class III) [26,27], whose hallmark lesion on a kidney biopsy is nodular glomerulosclerosis, or Kimmelstiel-Wilson nodules. Class IV is characterized by sclerosis in >50% of the glomeruli [28]. These patients often have ESRD.

Regarding non-albuminuric DKD, in 2019, Yamanouchi et al. [29] retrospectively assessed 526 patients with DKD, showing that normal or almost normal renal structure was most common (62%) in the patients with non-albuminuric DKD, while the typical DKD pattern was most prevalent (66%) in the albuminuric DKD group.

Accordingly, in a study by Ekinci et al. [30], it was found that the characteristic glomerular alterations of DKD were less frequent in non-albuminuric patients than in patients with micro- or macro-albuminuria. This showed multifactorial pathophysiology for the renal disease in these patients, with potential contributions from aging, hypertension, and vascular disease. The characteristic glomerular alterations of DKD were seen in virtually all (22 of 23) patients with micro- or macro-albuminuria. Furthermore, patients with normo-, micro-, and macro-albuminuria had mesangial areas that were a little larger than those of participants without DKD. Given that mesangial expansion and GBM thickness are correlated with lower eGFR in individuals with albuminuric DKD [31], this result was not surprising. This supported the idea that mesangial growth and eGFR reduction in DKD are connected [25]. By contrast, in the non-albuminuric phenotype, these histological changes were seen less frequently, with major tubulointerstitial and vascular (with varying degrees of arteriosclerosis) rather than glomerular involvement, suggesting a different pathogenetic process in this phenotype [30].

### 3.2. Pathogenesis

The pathogenesis of non-albuminuric DKD is still unknown, and in recent years, various hypotheses have been formulated to explain the pathogenesis of this phenotype.

There is growing evidence linking the development of the non-albuminuric phenotype in patients with diabetes to acute kidney injury (AKI) to CKD transition. The most likely hypothesis is the development of small and repeated episodes of AKI, sometimes even subclinical, of any nature, ischemic, infectious, toxic, or obstructive. This determines the development of CKD due to tubular damage. Diabetic subjects are more exposed to this type of damage for the following reasons: the greater tendency to tubular hypoxia [32], being in therapy with RAAS blockers that increase the susceptibility of the tubule to renal hypoxia [33], and a lower capacity for tubular regeneration [34,35].

Patients with diabetes often suffer multiple episodes of AKI due to vascular changes, endothelial cell injury, toxicity associated with medications, and multiple surgeries, while some episodes of mild AKI may go undetected. Thus, episodes of AKI in patients with diabetes are likely responsible for the DKD transition. It is also likely that the AKI-to-CKD transition is responsible for the decline of eGFR in patients with non-albuminuric DKD, which is characterized mostly by tubulointerstitial injury and fibrosis [36].

Moreover, it has been suggested that non-albuminuric DKD probably underlies macroangiopathy instead of microangiopathy as the prevailing pathology; the weak association of the non-albuminuric phenotype with diabetic retinopathy seems to confirm this statement [13,37]. Conversely, a recent study in patients with type 2 diabetes, reduced eGFR, and various degrees of albuminuria showed that, while typical glomerulopathy was observed in virtually all subjects with micro- or macro-albuminuria, only half of the normoalbuminuric patients had typical lesions and almost all of them had varying degrees of arteriosclerosis [30].

Few patients with diabetes and decreased eGFR in the absence of proteinuria have been biopsied and the results reported. Any number of other renal lesions could be present in these patients, including atheroembolism, renovascular disease, or tubulointerstitial disease from the many medications used to treat comorbidities. Lastly, patients with diabetes have a high risk for cardiovascular events and many comorbidities that confer risk for AKI. It is possible that unresolved episodes of AKI account for the decreased eGFR seen in many non-proteinuric patients with diabetes. Lastly, non-proteinuric diabetic kidney disease may represent a genetically different form of DKD.

## 4. Cardiovascular Risk Profile in DKD Phenotypes

Increased urinary albumin excretion and reduced eGFR have both been demonstrated to be risk factors for cardiovascular disease [38]. The more severe the renal impairment, or the higher the albuminuria, the greater the risk of cardiovascular as well as other complications. Patients with alteration of glucose homeostasis and type 2 diabetes have higher cardiovascular (CV) risk [39]. However, the question of which of the two phenotypes is associated with a higher cardiovascular risk is the focus of this review. To date, the topic remains debated due to contrasting results in the available literature.

The Second Nord-Trøndelag Health (HUNT II) study reported that the presence of microalbuminuria and reduced eGFR was associated with a higher risk for cardiovascular death in 9709 community-based participants [40]. The results of a nationwide observational study from the Swedish National Diabetes Register (66,065 patients with T2D, with a follow-up of 5.7 years) also confirmed this concept [41]. Similarly, in a study conducted on patients with type 1 diabetes, the risk of all-cause death was similar between reduced eGFR and albuminuria alone, with the highest mortality observed in patients presenting with both reduced eGFR and albuminuria [42].

In a post hoc analysis of the Action in Diabetes and Vascular Disease Preterax and Diamicron-MR Controlled Evaluation (ADVANCE) study [38], the effect of albuminuria and reduced eGFR on a total of 10,640 patients was investigated. On an average of 4.3 years follow-up, 938 (8.8%) patients experienced a cardiovascular event, 432 (4.1%) of which were fatal. Higher UACR levels and lower eGFR levels were both independently log-linearly associated with an increased risk for cardiovascular events and death, after adjustment for other cardiovascular risk factors. Patients with both UACR > 300 mg/g and eGFR < 60 mL/min/1.73 m^2^ had a 3.2-fold higher risk for cardiovascular events compared with patients with neither of these risk factors. Furthermore, the risk for cardiovascular events was lower in patients with normoalbuminuria and stage 3 CKD than in those with albuminuria and stage 2 CKD, suggesting the need for an additional stratification of cardiovascular risk on the basis of the presence or absence of albuminuria. Conversely, a post hoc analysis of the Fenofibrate Intervention and Event Lowering in Diabetes (FIELD) study (9795 participants with type 2 diabetes) [43] showed that the non-albuminuric phenotype was associated with a higher risk of cardiovascular events and cardiovascular death, compared to microalbuminuria with an eGFR >60 mL/min/1.73 m^2^ and macroalbuminuria with an eGFR > 90 mL/min/1.73 m^2^. In the RIACE trial [44], 15,773 individuals with type 2 diabetes were enrolled and classified based on albuminuria and eGFR in four groups: no DKD (Alb−/eGFR−), albuminuria alone (Alb+/eGFR−), reduced eGFR alone (Alb−/eGFR+), or both albuminuria and reduced eGFR (Alb+/eGFR+). All-cause mortality risk was lowest for Alb−/eGFR− and highest for Alb+/eGFR+, with similar values for Alb+/eGFR− and Alb−/eGFR+. However, the all-cause mortality risk was higher in the non-albuminuric group with eGFR < 45 mL/min/1.73 m^2^ (Alb−/eGFR+) than in the group with microalbuminuria alone (Alb+/eGFR−).

In a large cohort of Japanese diabetic patients, Yokohama et al. [45] investigated the risk of death, cardiovascular events, and decline of renal function according to DKD phenotypes. They showed that the risk of these outcomes was not higher in non-albuminuric DKD compared with other DKD phenotypes. Particularly, the risk was relatively low in non-albuminuric DKD without prior CV disease and was similar to those with no-DKD without prior CV disease. These findings, therefore, indicate that non-albuminuric DKD could reflect a unique phenotype with macroangiopathy as the underlying pathogenesis, with a subsequent impact on the prognosis of CV events and renal dysfunction. In non-albuminuric DKD, the occurrence of macrovascular problems may be a more important prognostic factor rather than renal manifestations.

In the Jin et al. [46] trial, the risks of CV and renal outcomes among patients with various DKD phenotypes were compared. It was found that both patients with non-albuminuric DKD (Alb−/eGFR+) and albuminuric DKD (Alb+/eGFR−) had a higher risk of hospitalization for heart failure. As concerns CV disease, the authors found that subjects with non-albuminuric DKD (Alb−/eGFR+) did not have a significant excess risk in comparison with no-DKD subjects, while patients with albuminuric DKD (Alb+/eGFR−) present a higher risk than no DKD.

In Table 2 we summarize data from the previously presented studies.

In light of all these results, it remains uncertain which DKD phenotype presents a higher cardiovascular risk, requiring further study and meta-analysis to clarify this debate. The same is true about the link between microalbuminuria and cardiovascular risk, which still remains poorly understood. It has been suggested that a common pathophysiologic process could underly this association, such as endothelial dysfunction, chronic low-grade inflammation, or increased transvascular leakage of macromolecules [47]. In the study by Bigazzi et al., it was demonstrated that hypertensive patients with microalbuminuria had an increased thickness of the carotid intima and media layers suggesting a greater degree of atherosclerosis [48]. A similar hypothesis has been formulated about the common pathogenetic process underlying the association between reduced eGFR and increased cardiovascular risk, suggesting that reduced eGFR may simply represent the renal manifestations of systemic atherosclerosis [49].

## 5. Risk of Renal Progression in DKD Phenotypes

DKD is the leading cause of CKD and ESRD globally, contributing to half of new kidney failure cases [2].

Whether diabetic patients with non-albuminuric DKD have a higher risk of kidney function deterioration and ESRD progression and mortality than those with albuminuric DKD is still an unmet clinical need; to date, several studies have addressed this specific issue.

In the Jin et al. trial [46], renal outcomes were also investigated. It was found that the risk of CKD progression was higher in patients with albuminuria with or without decreased eGFR compared with those with decreased eGFR without albuminuria. Similarly, Yokohama et al. [45] showed that the annual decline rate in eGFR was slower in non-albuminuric DKD than in albuminuric DKD. Kidney outcomes have also been investigated in a multicenter prospective cohort study conducted on 19,025 Chinese patients with type 2 diabetes enrolled in the Hong Kong Diabetes Biobank; the authors showed a higher risk of CKD progression in patients with decreased eGFR without albuminuria compared with those with no DKD, but lower incidence than those with albuminuria [46]. The Chronic Renal Insufficiency Cohort (CRIC) study [50] was conducted in patients with diabetes and CKD and demonstrated that those with non-albuminuric DKD have a much lower risk for ESRD, CKD progression, or rapid decline in eGFR compared with those in whom albuminuria or proteinuria is present. This low rate of eGFR decline in non-albuminuric DKD is consistent with the analysis conducted by Buyadaa et al. in 2020 [51].

According to the results reported above, patients with albuminuric DKD may have a higher risk of DKD progression and ESRD incidence compared with those with non-albuminuric DKD. Further studies are needed to confirm these data.

## 6. Therapy

Treatment for DKD, its comorbidities, and complications is challenging; as a result, a multifactorial strategy is needed.

Lifestyle modifications such as weight loss, increased physical activity, smoking cessation, Mediterranean diet and sodium restriction are the first step in DKD management; then, a pharmacological approach with lipid-lowering treatment [52] and blood pressure control is needed [53].

Until recently, the only drugs with evidence of benefit on DKD were Angiotensin-Converting Enzyme (ACE) inhibitors and Angiotensin Receptor Blockers (ARBs) because of their effect of albuminuria reduction or regression and slowing of CKD progression. However, over the past few years, much evidence has accumulated on other drug classes that have shown a nephroprotective effect in diabetic patients. Among the drug classes involved: Mineralocorticoid Receptor Antagonists (MRAs) and glucose-lowering agents, including glucagon-like peptide 1 receptor agonist (GLP1-RA) and sodium/glucose co-transporter 2 (SGLT2) inhibitors. However, none of these drug classes has yet been tested in large studies focused on the non-albuminuric DKD phenotype.

We will outline below the existing evidence on the various drug classes in DKD in general and, afterwards, the limited evidence on the non-albuminuric DKD phenotype, drawn from sub-analyses of large trials conducted in recent years.

### 6.1. Angiotensin-Converting Enzyme (ACE) Inhibitors or Angiotensin Receptor Blockers (ARBs)

ACE inhibitors or ARBs are the preferred first-line agent for hypertension treatment among patients with diabetes and UACR ≥300 mg/g because of their proven benefits for the prevention of CKD progression [54]. Lewis et al. demonstrated that captopril significantly slowed the rate of renal function loss in patients with DKD and urinary protein excretion ≥ 500 mg per day [55]. A multiple linear regression analysis confirmed that ACE inhibitors decrease proteinuria independently of blood pressure, treatment duration, type of diabetes or stage of nephropathy [56]. In the Reduction of Endpoints in NIDDM with the Angiotensin II Antagonist Losartan (RENAAL) study [57], in a cohort of diabetic patients with UACR > 300 mg/g or proteinuria > 500 mg/die, it was shown that losartan reduced proteinuria, which was the strongest risk factor for kidney failure. However, this study did not take into account patients without albuminuria/proteinuria but with a reduced eGFR (non-albuminuric phenotype). Even Brenner et al. found that losartan conferred significant renal benefits in patients with diabetes and nephropathy, defined as UACR > 300 mg/g [58].

Conversely, in the setting of lower levels of albuminuria (30–299 mg/g), ACE inhibitors or ARBs therapy has been shown to reduce progression to more advanced albuminuria (≥300 mg/g) and cardiovascular events but not progression to kidney failure [59]. Based on this evidence, current guidelines recommend the use of ACE inhibitors and ARBs for diabetic patients with albuminuria. Whether these drugs can be used for non-albuminuric DKD treatment is currently unclear since no trials have been conducted specifically on patients with eGFR reduction alone (without albuminuria).

### 6.2. Mineralocorticoid Receptor Antagonists (MRAs)

MRAs also significantly reduce albuminuria when added to ACE inhibitors or ARBs, but their use is limited by the risk of hyperkalemia, especially in patients with renal impairment [60].

Finerenone has been shown to reduce the UACR in patients with DKD treated with a renin-angiotensin system (RAS) blocker (i.e., ACE inhibitors or ARBs) while having smaller effects on serum potassium levels than other MRAs [61]. The Finerenone in Reducing Kidney Failure and Disease Progression in Diabetic Kidney Disease (FIDELIO-DKD) trial [62] was conducted in patients who had a UACR of 30 mg/g to less than 300 mg/g, an eGFR of 25 to less than 60 mL/min/1.73 m^2^, and diabetic retinopathy, or a UACR of 300 mg/g to 5000 mg/g and an eGFR of 25 to less than 75 mL/min/1.73 m^2^. This trial demonstrated that treatment with finerenone resulted in a lower risk of CKD progression (i.e., a decrease of ≥40% in the eGFR from baseline, or death from renal causes) and cardiovascular events (i.e., death from cardiovascular causes, nonfatal myocardial infarction, nonfatal stroke, or hospitalization after heart failure) than a placebo. The Cardiovascular Events with Finerenone in Kidney Disease and Type 2 Diabetes (FIGARO-DKD) trial [63] extended the findings of the FIDELIO-DKD trial to a wider spectrum of DKD patients (UACR of 30 mg/g to less than 300 mg/g and eGFR of 25 to 90 mL/min/1.73 m^2^ or UACR of 300 mg/g to 5000 mg/g and eGFR of at least 60 mL/min/1.73 m^2^) focusing on a primary cardiovascular outcome, demonstrating the protective CV effect of finerenone in these patients.

A FIDELITY [Finerenone in Chronic Kidney Disease and Type 2 Diabetes: Combined FIDELIO-DKD and FIGARO-DKD Trial Programme Analysis] analysis of data from both trials (FIDELIO-DKD and FIGARO-DKD) [64] concluded that finerenone improved both cardiovascular and kidney outcomes across the spectrum of CKD severity stages. Results from the FIDELITY sub-analysis added the improvement of heart failure (HF) related outcomes in DKD to the benefits of treatment with finerenone, regardless of eGFR and/or UACR categories. However, this study, like the previous ones, enrolled patients with at least microalbuminuria (UACR > 30 mg/g), excluding patients with only eGFR reduction.

In a recent network meta-analysis [65], a total of 12 randomized clinical trials with 15,492 patients applying various types of MRAs covering spironolactone, eplerenone, finerenone, esaxerenone, and apararenone were included. Several MRAs significantly reduced the UACR in patients with DKD, whereas, regarding the change in eGFR, finerenone may have potential superiority in protecting the kidney.

### 6.3. Glucose-Lowering Agents

Intensive glycemic control has been shown in large randomized clinical trials to delay the onset and progression of DKD in diabetic patients [66]. The choice of an antidiabetic regimen for patients with type 2 diabetes relies on an interplay of patient characteristics, severity of hyperglycemia, and available therapeutic options. Pharmacological interventions used to improve glucose control include both oral and injectable drugs.

Several glucose-lowering agents, such as metformin, DiPeptidyl Peptidase 4 (DPP-4) inhibitors [67], GLP1-RA [68], thiazolidinediones [69], and SGLT2 inhibitors [70] were demonstrated to have a pleiotropic renoprotective action [71].

Metformin is the first-choice antidiabetic medication for most cases. Christiansen et al. showed that patients who started metformin reported a lower risk of experiencing a severe decline in eGFR [72]. There are conflicting results about the impact of metformin-associated lactic acidosis on the risk of developing acute kidney injury. Mariano et al. reported that the use of metformin was associated with the occurrence of lactic acidosis, resulting in AKI that required dialysis [73]. On the contrary, the study by Bell et al. [74] found no significant association of metformin use with an increased risk for AKI in terms of lactic acidosis.

De Bhailís et al. [75] in a review focused on the role of GLP1-RAs and SGLT-2 inhibitors in treating patients with DKD and clearly demonstrated that these agents should now be considered the first choice in combination with metformin in patients with diabetes and increased cardiovascular risk and/or reduced renal function, in preference to other classes such as DPP-4 inhibitors or sulphonylureas.

#### 6.3.1. DPP-4 Inhibitors

Four cardiovascular outcome trials (CVOTs) evaluated the CV and renal safety of DPP-4 inhibitors in a total of 43,522 patients. In the SAVOR-TIMI 53 trial [76], type 2 diabetes patients with either a history of established CV disease or multiple risk factors for CV disease randomly received saxagliptin or a placebo. There was a significant difference in the saxagliptin arm, as compared with a placebo, in the progression to macroalbuminuria. However, saxagliptin did not cause any significant change in the composite renal outcome, nor in eGFR, doubling of serum creatinine, initiation of chronic dialysis and renal transplantation. The Alogliptin after Acute Coronary Syndrome in Patients with Type 2 Diabetes (EXAMINE), Trial Evaluating Cardiovascular Outcomes with Sitagliptin (TACOS) and Cardiovascular and Renal Microvascular Outcome Study With Linagliptin (CARMELINA) trials [77,78,79], on alogliptin, sitagliptin, and linagliptin, respectively, also produced similar results, with no statistically significant differences on CV and DKD outcomes. In light of these trials, it can be deduced that DPP-4 inhibitors did not show a positive effect on renal outcomes.

#### 6.3.2. GLP1-RAs

Many studies showed the potential renoprotective effects of GLP1-RAs. In a secondary analysis of the Liraglutide Effect and Action in Diabetes: Evaluation of Cardiovascular Outcome Results (LEADER) trial [80], Liraglutide has been reported to maintain the impact on reducing macroalbuminuria over approximately four years. Dulaglutide reduced the composite renal endpoint and had a significantly lower incidence rate of macroalbuminuria than the placebo in the Dulaglutide and cardiovascular outcome in type 2 diabetes (REWIND) trial [81]. Dulaglutide also reduced the decline of eGFR following 52 weeks of therapy in the study by Tuttle et al. [82].

Similar positive findings were reported with another two GLP1-RAs; semaglutide [83] and exenatide [84], in which both drugs reduced the incidence and progression of DKD in patients with diabetes.

#### 6.3.3. SGLT2 Inhibitors

Data are available in the literature about the important renal protection effect exerted by SGLT2 inhibitors on DKD, which act with different mechanisms of action, as shown in Figure 2. There is not only a reduction or regression of albuminuria, but also a slowing down of the doubling time of creatininemia and therefore of the reduction of the eGFR, and ultimately the slowing of progression to ESRD and dialysis. This is a class effect.

SGLT2i exerts indirect nephroprotective effects through the improvement of HBA1c, reduction of body weight, reduction of glucotoxicity, improvement of insulin resistance, and of pancreatic beta-cell function. They also stimulate osmotic diuresis and natriuresis, with a reduction of plasma volume, anti-edema effects, and, therefore, reduction of blood pressure [86]. Moreover, it seems that they also have an effect on the reduction of arterial stiffness, not only through the reduction of blood pressure values but also with the increase in the urinary excretion of uric acid [85,87]. By inhibiting the Na/H exchanger at the endothelial level, they reduce intracellular calcium concentrations, resulting in an increase in nitric oxide (NO), with consequent vasodilation. All these effects indirectly reduce risk factors for DKD progression, slowing down this process.

Direct nephroprotective effects of SGLT2i have also been proposed: the first and most important is the action on the hemodynamic pathway, through afferent arteriolar vasoconstriction, thereby suppressing intraglomerular pressure and therefore reducing glomerular hyperfiltration [88]; downregulation of the RAS; reduction of proinflammatory and profibrotic factors [89]; and finally, the effect of increased bioavailability of oxygen to the kidney, and therefore reduction of hypoxia and ischemia [90].

The systematic review and meta-analysis of trials on SGLT2i by Neuen et al. [91] confirmed that this class of drugs reduces the risk of dialysis, kidney transplantation, and renal death in patients with DKD and also provides protection against AKI.

Three large multinational, randomized trials that were primarily designed to evaluate the cardiovascular outcomes of dapagliflozin (Dapagliflozin Effect on Cardiovascular Events, DECLARE–TIMI 58), canagliflozin (Canagliflozin and Cardiovascular and Renal Events in Type 2 Diabetes, CANVAS Program), and empagliflozin (Empagliflozin, Cardiovascular Outcomes and Mortality in Type 2 Diabetes, EMPA-REG OUTCOME) reported the renal endpoints as secondary outcomes. The Canagliflozin and Renal Outcomes in Type 2 Diabetes and Nephropathy (CREDENCE) trial, instead, was primarily designed to assess the renal safety of canagliflozin in diabetic patients with pre-existing renal disease.

The DECLARE-TIMI 58 study [92], which secondarily examined the renal effects of dapagliflozin in diabetic patients with cardiovascular disease or at high risk for cardiovascular disease (*n* = 17,160), provided the strongest evidence. Participants on dapagliflozin had a lower risk of renal mortality or ESRD as well as a reduced risk of a sustained decline in eGFR. This positive nephroprotective effect was shown in a similar population for canagliflozin from the CANVAS Program study (*n* = 10,142) [93] and empagliflozin from a secondary analysis of the EMPA-REG OUTCOME study among Asian patients (*n* = 1517) [94]. The CREDENCE trial (*n* = 4401) also reported reductions in the proportions of participants using canagliflozin and developing ESRD, dialysis, or renal death [95].

Additionally, in the Empagliflozin Outcome Trial in Patients With Chronic Heart Failure and a Reduced Ejection Fraction (EMPEROR-Reduced) trial [96], the rate of decline of eGFR over the duration of the treatment period was slower in the empagliflozin group than in the placebo group, with a significant difference of 1.73 mL per minute per 1.73 m^2^ per year between the study groups.

### 6.4. Non-Albuminuric DKD Therapy

Currently, there are no specific therapies for non-albuminuric DKD. It is generally assumed that management of several risk factors such as high glucose levels, high blood pressure, hypercholesterolemia, and other factors, can protect renal function and delay the progression to chronic renal failure; however, whether this approach may be effective in patients with non-albuminuric DKD remains to be shown.

Whether RAS blockers can be used for non-albuminuric DKD treatment is currently unclear. According to Dwyer et al., RAS inhibitors may slow the progression of non-albuminuric DKD by reducing proteinuria, but more studies are required to confirm this hypothesis since there are still subjects whose proteinuria goes down but nonetheless have a progression of their underlying renal status [97].

The American Diabetes Association and the European Association for the Study of Diabetes (ADA/EASD) published an updated Consensus Report for the management of hyperglycemia in diabetics in 2022 [98] recommending metformin in association with lifestyle measures (weight loss and physical activity) as a first-line treatment, then distinguishing between patients with established DKD, heart failure, or atherosclerotic cardiovascular disease (ASCVD). Recent randomized controlled trials (RCTs) demonstrated that both SGLT-2i and GLP1-RAs reduce cardiovascular and renal events in type 2 diabetes patients [99]. Based on growing clinical evidence, the American Diabetes Association recommends SGLT2i or GLP1-RAs in patients with ASCVD or DKD. However, no head-to-head RCT has directly compared the risk-reduction effects of SGLT2i and GLP1-RAs on cardiovascular and renal events in diabetic patients according to the presence or absence of albuminuria as the primary outcome. Accordingly, data for patients with this phenotype should be extrapolated from subgroup analysis of CVOTs, including patients with and without albuminuria [100].

These classes of drugs have extensively demonstrated their protective effect on kidney disease in diabetic patients, as previously reported. However, data concerning therapy in the non-albuminuric phenotype are still inadequate and no studies have been carried out investigating this subgroup of patients with DKD. Nonetheless, some data come from sub-analyses of important studies on SGLT2i.

The Empagliflozin in Patients with Chronic Kidney Disease (EMPA-KIDNEY) trial [101] was designed to investigate whether empagliflozin reduces the risk of kidney disease progression or CV death in patients with CKD. The population enrolled had a broad range of eGFR and was divided according to eGFR and the presence/absence of proteinuria UACR. On a total of 6609 patients, progression of kidney disease or CV death occurred in 432 out of 3304 patients (13.1%) in the empagliflozin group and in 558 out of 3305 patients (16.9%) in the placebo group, with consistent results among patients with or without diabetes and across subgroups defined according to eGFR ranges and UACR.

In the Dapagliflozin in Patients with Chronic Kidney Disease (DAPA-CKD) study [102], adults with or without type 2 diabetes who had an eGFR of 25 to 75 mL/min/1.73 m^2^ and a UACR of 200 mg/g to 5000 mg/g were randomly assigned to receive dapagliflozin or a placebo. Among patients with CKD, regardless of the presence or absence of diabetes, the risk of a composite of a decline in the eGFR of at least 50%, ESRD, or renal or CV death was significantly lower with dapagliflozin than with a placebo.

The CANVAS-Renal trial [93] demonstrated the benefit of canagliflozin in diabetic patients with high CV risk with respect to albuminuria progression and eGFR reduction rate, risk of ESRD, dialysis, and renal death.

However, none of these studies specifically analyzes the distinction of the effect of SGLT2i on the various phenotypes of DKD. Unfortunately, subgroup analysis evaluated patients only on the basis of preserved/reduced eGFR and presence/absence of albuminuria, without considering both factors simultaneously for defining the phenotypes, albuminuric and non-albuminuric, as we know them now. This prevented them from drawing any conclusion on the possible therapeutic effect of these drugs in non-albuminuric DKD.

For this reason, there are still no specific guidelines on the use of these drugs in one phenotype rather than the other, generically referring to diabetic patients with CKD. In the future, further studies investigating the CV and renal risk in diabetic patients with renal disease will have to be conducted on the various classes of drugs, especially investigating the difference between the classic form of DKD with albuminuria and the new non-albuminuric form with reduction of glomerular filtrate.

## Figures and Tables

**Figure 1 biomolecules-13-00752-f001:**
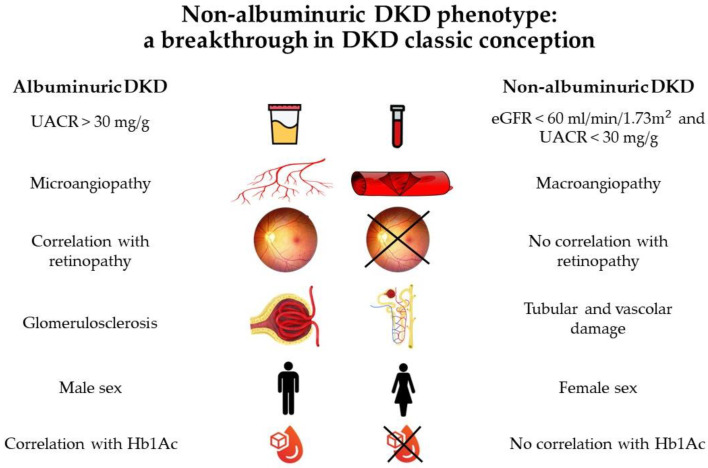
Differences between albuminuric and non-albuminuric phenotypes: diagnosis [5], clinical presentation [7], risk factors [13,14], and pathogenesis [16].

**Figure 2 biomolecules-13-00752-f002:**
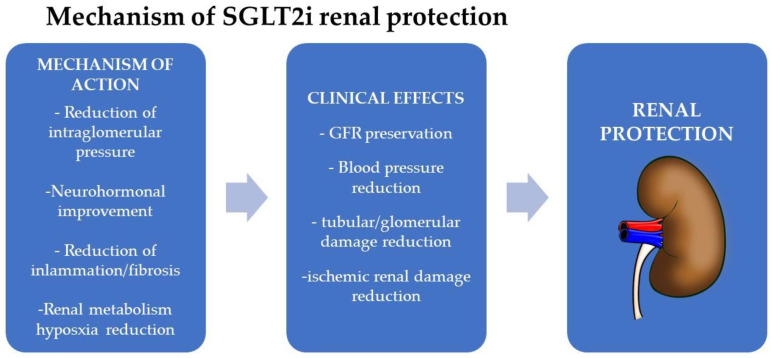
Mechanisms of SGLT2i renal protection. Adapted from Leoncini G. et al., 2021 [85].

**Table 1 biomolecules-13-00752-t001:** Histological manifestations of diabetic kidney disease.

Class	Description
I	Mild or nonspecific light microscopy changes and electron microscopy-proven GBM thickening [24]
IIa	Mild mesangial expansion (in >25% of the observed mesangium) [25]
IIb	Severe mesangial expansion (in >25% of the observed mesangium)
III	Nodular sclerosis (Kimmelstiel-Wilson lesion) [26,27]
IV	Advanced diabetic glomerulosclerosis (<50% of glomeruli) [28]

Adapted from Tervaert et al., 2010 [23]. GBM: glomerular basement membrane.

**Table 2 biomolecules-13-00752-t002:** Studies on DKD phenotypes and cardiovascular risk.

ADVANCE Post-Hoc Analysis (Ninomiya et al., 2009) [38] *

**Cardiovascular events**			
	***eGFR* ≥ *90***	***eGFR 60*–*89***	***eGFR* < *60***
** *Normoalbuminuria* **	1.00 (reference)	0.98 (0.78–1.22)	1.33 (1.02–1.75)
** *Microalbuminuria* **	1.48 (1.09–2.01)	1.54 (1.20–1.98)	2.04 (1.54–2.69)
** *Macroalbuminuria* **	1.18 (0.52–2.69)	1.67 (1.09–2.57)	3.23 (2.20–4.73)


**Cardiovascular death**			
	***eGFR* ≥ *90***	***eGFR 60*–*89***	***eGFR* < *60***
** *Normoalbuminura* **	1.00 (reference)	1.22 (0.81–1.84)	1.85 (1.17–2.92)
** *Microalbuminuria* **	1.96 (1.16–3.32)	2.52 (1.65–3.84)	3.37 (2.15–5.30)
** *Macroalbuminuria* **	2.87 (1.01–8.18)	3.61 (2.02–6.43)	5.93 (3.45–10.20)

**FIELD posthoc Analysis (Drury et al., 2011)** [43] **^#^**


**Cardiovascular** **events**	
	***eGFR* ≥ *90***	***eGFR 60*–*89***	***eGFR* < *60***
** *Normoalbuminura* **	1.00 (reference)	1.11 (0.95–1.29)	1.63 (1.20–2.20)
** *Microalbuminuria* **	1.25 (1.01–1.54)	1.43 (1.18–1.72)	1.94 (1.37–2.73)
** *Macroalbuminuria* **	1.19 (0.76–1.85)	1.77 (1.33–2.36)	2.30 (1.48–3.55)

**Cardiovascular death**			
	***eGFR* ≥ *90***	***eGFR 60*–*89***	***eGFR* < *60***
** *Normoalbuminura* **	1.00 (reference)	1.17 (0.80–1.72)	2.36 (1.29–4.31)
** *Microalbuminuria* **	1.73 (1.08–2.77)	1.38 (0.88–2.15)	2.96 (1.59–5.51)
** *Macroalbuminuria* **	1.89 (0.83–4.27)	2.59 (1.49–4.50)	5.26 (2.73–10.15)




**RIACE (Penno et al., 2018)** [44] **^@^**

**All-cause death**
	** *eGFR* ** ** *≥90* **	** *eGFR* ** ***75*–*89***	** *eGFR* ** ***60*–*74***	** *eGFR* ** ***45*–*59***	** *eGFR* ** ***30*–*44***	** *eGFR* ** ** *<30* **
** *UACR* ** ** *<10* **	1.00(ref.)	0.80(0.67–0.96)	1.10(0.83–1.12)	1.32(1.97–1.62)	1.85(1.40–2.44)	1.61(0.88–2.97)
** *UACR* ** ***10*–*29***	0.94(0.78–1.12)	1.05(0.89–1.25)	1.06(0.88–1.27)	1.39(1.14–1.69)	2.25(1.79–2.82)	2.25(1.49–3.37)
** *UACR* ** ***30*–*299***	1.31(1.08–1.60)	1.31(1.09–1.58)	1.39(1.15–1.68)	1.48(1.22–1.80)	2.09(1.69–2.59)	2.79(2.09–3.70)
** *UACR* ** **≥*300***	2.19(1.55–3.11)	2.48 (1.82–3.38)	1.71(1.23–2.36)	2.26(1.71–3.00)	2.78(2.14–3.63)	4.66(3.59–6.05)

**JDDM 54 (Yokoyama et al., 2020)** [45] **^$^**


	***Alb***− ***eGFR***−	***Alb*+ *eGFR***−	***Alb***− ***eGFR*+**	***Alb*+ *eGFR*+**
** *CVD* **	1.00 (reference)	1.75 (1.32–2.34)	1.06 (0.63–1.79)	2.30 (1.57–3.39)
** *Death or CVD* **	1.00 (reference)	1.73 (1.35–2.21)	1.02 (0.66–1.60)	2.32 (1.67–3.24)

**Analysis from Hong Kong Diabetes Biobank (Jin et al., 2022)** [46] **°**

	***Alb***− ***GFR***−	***Alb*+ *GFR***−	***Alb***− ***GFR*+**	***Alb*+ *GFR*+**
***All*-*cause mortality***	1.00 (reference)	2.00 (1.52–2.63)	1.59 (1.04–2.44)	3.26 (2.43–4.38)
** *CVD* **	1.00 (reference)	1.19 (1.02–1.40)	1.14 (0.88–1.48)	1.47 (1.23–1.76)
** *Hospitalization for HF* **	1.00 (reference)	3.14 (2.09–4.73)	3.08 (1.82–5.21)	5.50 (3.63–8.34)


* Values are expressed as Hazard Ratio (HR) (95% CI), adjusted for age, sex, diabetes duration, systolic blood pressure (BP), anti-hypertensive treatment, history of macrovascular disease, HbA1c, HDL-cholesterol, log-transformed triacylglycerol, body mass index (BMI), electrocardiogram abnormalities, smoking status, and drinking status. eGFR: estimated glomerular filtration rate. ^#^ Values are expressed as HR (95% CI) adjusted for age, sex, diabetes duration, smoking status, BMI, systolic BP, HbA1c, HDL-cholesterol, LDL-cholesterol, triacylglycerol, retinopathy, and renin-angiotensin-aldosterone inhibition. eGFR: estimated glomerular filtration rate. ^@^ Values are expressed as HR (95% CI) adjusted for age, sex, smoking status, diabetes duration, HbA1c, BMI, estimated waist circumference, triacylglycerols, total cholesterol, HDL-cholesterol, lipid-lowering treatment, systolic and diastolic BP, anti-hypertensive treatment, diabetic retinopathy grade, any cardiovascular disease, and any cancer. UACR: urinary albumin to creatinine ratio. eGFR: estimated glomerular filtration rate; ref: reference. ^$^ Values are expressed as HR (95% CI), adjusted for age, sex, diabetes duration, BMI, smoking status, HbA1c, systolic BP, anti-hypertensive treatment, HDL and non-HDL cholesterol, and lipid-lowering treatment. CVD: cardiovascular disease. Alb− GFR−: no diabetic kidney disease (DKD); Alb+ GFR−: albuminuric DKD without reduced estimated Glomerular filtration rate (eGFR); Alb-GFR+ non-albuminuric DKD; Alb+ GFR+: albuminuric DKD with reduced eGFR.° Values are expressed as HR (95% CI), adjusted for age, sex, diabetes duration, BMI, smoking at any time, HbA1c, systolic BP, LDL cholesterol, HDL cholesterol, triacylglycerols, oral antihyperglycemic drugs, insulin, lipid-lowering drugs, renin-angiotensin system blockers, diabetic retinopathy, and history of cardiovascular disease and congestive heart failure. CVD: cardiovascular disease. HF: heart failure. Alb− GFR−: no diabetic kidney disease (DKD); Alb+ GFR−: albuminuric DKD without reduced estimated Glomerular filtration rate (eGFR); Alb− GFR+ non-albuminuric DKD; Alb+ GFR+: albuminuric DKD with reduced eGFR. Colors: white, reference (no DKD) and HR with no statistically significant differences vs. no DKD; yellow, HR < 1.5 fold higher vs. no DKD in a statistically significant manner; red, HR >1.5 fold higher vs. no DKD in a statistically significant manner.

## Data Availability

Not applicable.

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
