# Peer review of "Update on Diabetic Kidney Disease (DKD): Focus on Non-Albuminuric DKD and Cardiovascular Risk"

_biomolecules, 2023, doi:10.3390/biom13050752_

Round 1

Reviewer 1 Report

The authors have carried out a comprehensive review updating readers on the current status of diabetic kidney disease with a unique focus on non-albuminuric DKD and its potential cardiovascular risk. The review is extensive and highlight the need to differentiate between albuminuric and non-albuminuric forms of DKD. However, I do have some concerns that I would like addressed.

1.    In table 1, the authors should add references corresponding to each phenotype listed.

2.    In a lot of places, the references are inserted at the end  of the paragraph and not when first stated. Slight editorial changes can be made to accommodate for ease of reading.

3.    In the section highlighting the role AKI-to-CKD transition in the pathogenesis of DKD, the authors cite another review article. Supplementing details from the original articles will help highlight this phenomenon much better.

4.    A summary table highlighting clinical trials for therapeutics can be created. Even listing if the drug/trial showed an impact on alb+ or alb- renal and/or CV risk would be beneficial to readers.

Minor comments.

1.    Line 307: is it  .. “>500 mg/dL”?

2.    Line 331: change spelling to “risk “

           3.    Line 387: change abbreviation DPP-4 

Reviewer 2 Report

This review provides a comprehensive summary of the recent advancements in DKD with a specific focus on the Non-albuminuric DKD. The review covers various aspects including clinical presentation, epidemiology, risk factors, pathogenesis, histopathology and different treatments. Overall, the review is clearly structured and well-written with an attractive topic that is of significance to readers.  However, there are several concerns that need to be addressed:

1: Subheadings should be added to part 2&3 to make them clearer and readable.

2: When talking about the prevalence of Non-albuminuric DKD, there is a systematic review that could be cited.

Front. Endocrinol., 03 June 2022

Volume 13 - 2022 | https://doi.org/10.3389/fendo.2022.871272

Comparison of Nonalbuminuric and Albuminuric Diabetic Kidney Disease Among Patients With Type 2 Diabetes: A Systematic Review and Meta-Analysis.

3: DKD is the main cause of CKD, ESRD and premature death. It would be more comprehensive to add the discussion of the risks of ESRD or all-cause death in Non-albuminuric DKD in addition to cardiovascular risks.

4: There are typos and grammar errors in the text that should be improved.

e.g. Line 57 “cOexisting”

5: Inappropriate use of abbreviations. All nonstandard abbreviations should be defined at first use in the text.

e.g. Line153 Use the full term of “chronic kidney disease” instead of the abbreviation, although it has been indicated earlier.

Overall, the review is clearly structured and well-written. Minor editing is required.
